# Hierarchical Fusion Process of Destination Image Formation: Targeting on Urban Tourism Destination

**Xuhui Zhang [1,2], Chen Zhang [1,2,*], Yanan Li [1,2], Ziyu Xu [1,2] and Zhenfang Huang [1,2]**

1   School of Geographical Science, Nanjing Normal University, Wenyuan Road, Qixia District, Nanjing 210023, China; kobezhang1996@126.com (X.Z.); lyn1446578487@126.com (Y.L.); xzynnuer@163.com (Z.X.); huangzhenfang@njnu.edu.cn (Z.H.)
2   Jiangsu Center for Collaborative Innovation in Geographical Information Resource Development and Application, Wenyuan Road, Qixia District, Nanjing 210023, China
*   Correspondence: zhangc@njnu.edu.cn; Tel.: +86-133-9090-8180

**Abstract:** Image has been widely accepted as a combination of perceived elements that are commonly discrete and static ones. 'Discrete' means that the elements are treated as separate ones with each other, with no interactions among them. 'Static' means that the elements would not be changed into other forms in the process of destination image formation. This study, thinking outside the box, tries to explore destination image formation through perceived elements and take their interactions and corresponding changes into account. Machine learning, as the core of artificial intelligence, is applied for data analysis in this study. Urban tourism destinations are targeted because of their variety and abundance of perceived elements. Data are collected from both interview and questionnaire surveys of tourists. Through several phases of analysis, this study finally finds that perceived elements do interact with each other and change into new forms level by level in tourism destination image formation. Specifically, there are four levels from bottom to top in the whole process of destination image formation, i.e., the individual-landscape layer, compound-atmosphere layer, dual-factor layer, and overall-image layer. In the bottom stage, elements are commonly numerous, separate, and concrete. With the interactive effects of the elements, they integrate with each other and generate some new forms in higher levels, which would be more general and abstract. Based on the findings, the dynamic fusion process and pyramid hierarchy of destination image formation are disclosed. This study explores destination image formation from a new perspective, considering perceived elements within a dynamic, synthetic system, and therefore provides practical insights into destination image construction in a more comprehensive and targeted way.

**Keywords:** destination image formation; perceived element; interactive effect; hierarchical structure; fusion process; urban tourism destination; machine learning

## 1. Introduction

Tourism destination image is one of the core concepts in tourism research. This concept is originally derived from the psychology field, and has been introduced and developed in multiple disciplines, such as biology, geography, and sociology. Scholars widely accept it as a mental representation of real-world objects [1], formed from people's perceptions, that strongly influences people's emotions and behavior toward that object [2–4].

Image formation is one of the fundamental constructs in tourism destination image research [5]. Scholars widely accept that destination image is composed of perceived elements referring to a particular place, such as buildings, residents, mountains, food, or public events [6–8]. Note that, though the interrelationship among the elements of destination image has been realized, they have always been explored in a discrete and static way. Here, 'discrete' means that these elements are separated from each other, with no interactive relationships among them, while 'static' means that the elements are invariable ones that would not be changed into other forms in the whole process of destination

image formation. Based on these views, the effects of the elements on a particular target (e.g., overall image or behavioral intention) have mostly been investigated in an isolated and single-layer way, with no interactive or hierarchical relationships being considered.

Actually, there are several issues that need to be considered: Are there interactions among perceived elements? Would the elements be changed, such as integrate or disintegrate, into some other forms in the interaction process? Additionally, is it possible that these interactions are in a dynamic hierarchical process? Based on these considerations, we attempt to explore destination image formation from a new perspective and take the interactions of perceived elements into account.

The interactions of perceived elements have been confirmed originally in the cognitive psychology field. Scholars found that the way people recognize a particular object is commonly composed of elements in multiple, interacting levels [9]. Specifically, people gain knowledge of an object through extensive, concrete elements first. With the accumulation of the elements in mind, they would be synthesized into relatively fewer abstract elements, level by level [10]. This is a hierarchical fusion process, with all the perceived elements within it being mutually associated and interacted. Actually, organizing lower-level elements into higher-order groupings and affording comprehensibility and flexibility to human behavior has been regarded as one of the hallmarks of the human mind [11].

Though the idea of hierarchical perception fusion has been applied in different domains, such as version, language, music, and spatial navigation [12–14], it has not been introduced into destination image research yet. Based on that, this study seeks to verify and explore the process of hierarchical perception fusion in destination image formation, with a focus on the interactions of perceived elements. An urban tourism destination is targeted in this study, mainly due to the variety and abundance of perceived elements referring to it.

Specifically, the aims of this study were three-fold: First, we wanted to confirm that there are interactions among perceived elements in destination image formation. Second, we needed to identify the hierarchical structure of interactions, including the layers and the elements within each layer. Third, we needed to explore the dynamic fusion process among different layers, through the interactions of elements. Based on these, the hierarchical fusion process of destination image formation would be finally disclosed. Machine learning methods, as the core of artificial intelligence, were applied in this study, so as to construct models that are capable of gracefully approaching the ground-truth relationships in the real world. The findings of this study could provide practical insights into brand image construction of urban tourism destinations in a comprehensive and targeted way.

## 2. Theoretical Background

### 2.1. The Conception of Destination Image

Early works on image embraced the view that it is a mental representation of objective reality residing and existing in the minds of humans [15]. Many works suggest that human behavior is much more dependent upon this mental representation rather than objective reality itself. This fact arises from academic interests in different fields and disciplines on image research. Hunt is the first to connect this mental representation to a place. He proposes the concept and destination image for tourism destination research. In his concept, destination image comprises the perceptions that people hold through cognition of a non-residence area. Perceptions here are the organizations, identifications and interpretations of a particular place through sensory information, such as vison, sound, touch, taste and smell. Crompton develops this concept by detailing perceptions as 'beliefs, ideas and impressions of a destination' [16]. Beliefs and ideas have been treated mostly as the objective estimation of the physical or functional attributes of a destination, whereas impressions commonly reflect the feelings of the entire place, which can be viewed as a more subjective estimation. This definition has been most widely accepted in tourism destination research up to now.

### 2.2. The Formation of Destination Image

There are classical models of destination image formation that have been widely adopted in tourism research. From a perspective of constituent elements, Echtner and Ritchie divide destination image into four parts, from three continuum dimensions (i.e., attributed-holistic, functional-psychological, common-unique) [17]; from a perspective of stimulus-response process, Baloglu and Mccleary indicate that destination image includes both cognitive and affective parts; additionally, from a perspective of information processing, Gunn finds that destination image is originated from an organic one, and then changed into induced and complex ones [18]. The model in this study was constructed from a new perspective, which has intersections with these models to a certain extent. We try to explain the relevance and differences of this model compared with previous classical ones, which are detailed as follows.

### 2.3. Destination Image from Parts to a Whole

Scholars describe destination image as a gestalt construct, with both individual and holistic components [19,20]. One typical example is the classical model, which divides destination image into attributed and holistic components through one of its major dimensions. Attributed components refer to individual attributes of a destination, with tangible or intangible forms, such as tall buildings, delicious food, cold weather, or friendly residents, while holistic components refer to the whole picture or characteristics of a destination with more general and abstract components, as described by Smith [21]. Based on gestalt principles, holistic components of destination image are actually generated from individual attributes that people perceive. This would not be simply equal to the sum of individual attributes together, but a regeneration from multi-perception fusion.

This model is one of the most influential works in destination image research. Though scholars ascribe equal importance to both individual and holistic components of destination image, more attention has been paid to individual ones, mainly due to their simplicity in description and measurement, whereas holistic ones have been relatively ignored. In this study, we try to explore destination image formation, with both individual and holistic components being considered. Rather than static structural relationships of destination image, we try to explore dynamic interaction-relationships between individual and holistic components.

### 2.4. Destination Image from Cognitive to Affective

Destination image can be divided into cognitive and affective parts, based on the subject's attitudinal reactions. Referring to cognitive image, some scholars indicate that this construct refers to physical or functional attributes of a destination, with respect to objective knowledge and beliefs that people may acquire [22,23], while some other scholars extend it into a larger scope, including cognition/perceptions of all aspects of a destination [24]. No matter which views are considered, cognitive image can be treated as a place-oriented construct. All the descriptions of it focus on 'place', with an induced statement: 'I feel that the destination is . . . '.

In terms of affective image, it seems that there is a divergence in referring to its definition. Some scholars treat it as a general feeling or impression that people have of a place. It means that it is still within a perceptual system, and is place-oriented; however, some other scholars indicate that affective image mainly refers to people's emotion or affection for a place [25–27]. A typical work is Russel's affective measurement model, which estimates affective image through four bipolar scales, i.e., Arousing-Sleepy, Pleasant-Unpleasant, Exciting-Gloomy, and Relaxing-Distressing. According to it, this construct is more people-oriented. All the descriptions of it focus on 'people' (to a certain extent), with an induced statement: 'This destination makes me feel . . . '.

We explore destination image from a place-oriented perspective, with all the elements of destination image being within a perceptual system. Actually, there are hierarchical relationships/structures referring to destination image that have been studied in previous

research, such as those from image and satisfaction/affection to behaviors. However, these elements in different layers are not in the same class, i.e., cannot be included within one system, and therefore should not be treated as a hierarchical fusion process, which is quite different from our study. We deconstruct destination image through perceived levels to break the boundary of cognitive and affective parts, trying to figure out the relationships among levels, as well as among elements within levels.

### 2.5. Destination Image from Organic to Complex

Reynolds indicates that the formation of image is based upon processing a flood of information that people acquire over time [28]. All sources and senses are involved in this processing course. Since human senses include sight, hearing, smell, taste, and touch, the information acquired from them is accordingly related to various types of perceived elements, with either tangible or intangible forms. Information sources include both first-hand and second-hand sources. Here, first-hand information is acquired directly through tourists' actual experiences on site, while second-hand information can be acquired through various channels, such as promotional literature, opinions of others, general media and network media [29]. With the accumulation of acquired information, destination image is formed correspondingly in a gradual changing process. One of the most influential works referring to it is Gunn's model, which investigates destination image through seven phases of travel experience. Through his model and following works based on it, destination image formation has been viewed as a process starting from an organic one, and then converted into an induced and complex one.

This study tries to explore the dynamic process of destination image formation as well. Note that both first-hand and second-hand channels are used in the information processing course. We pay more attention to the interaction and fusion process of destination image in a dynamic hierarchical way, compared with Gunn's model, which focuses on the changed images.

### 2.6. Conceptual Framework Development

We propose that destination image is formed in a dynamic hierarchical process, with perceived elements interacting with each other in stages. Specifically, in the bottom stage, the elements are separate, concrete, and commonly great in number. With the ascent of stages, these elements interact with each other, and fuse into some fewer abstract ones. The conceptual framework is illustrated as follows (Figure 1).

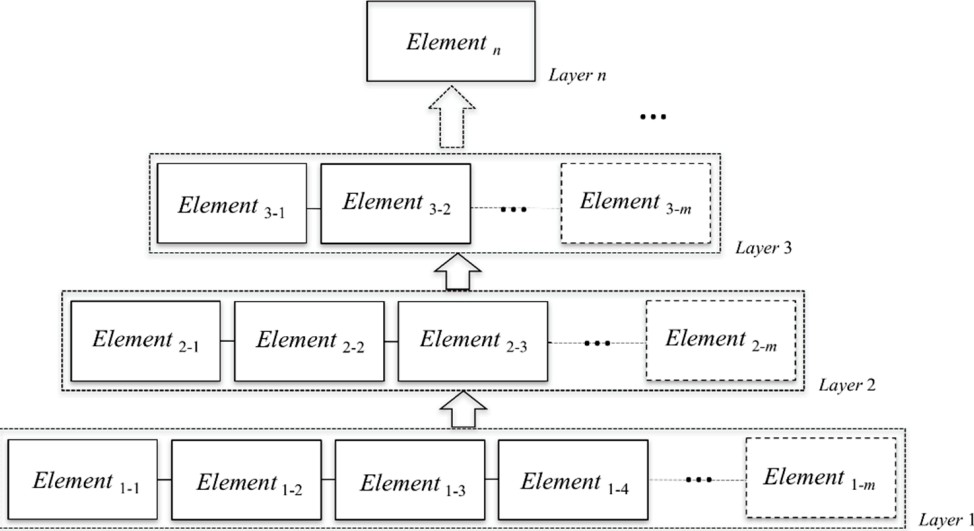

**Figure 1.** Conceptual framework of destination image formation in this study.

### 3. Methodology

Both interview and questionnaire surveys were applied in this study. An interview survey was used to preliminarily verify the existence of hierarchy in destination image formation, as well as to collect as many relevant perceived elements as possible. The questionnaire survey and subsequent data analysis were used to further explore the hieratical structure and dynamic process of destination image formation through interactive elements. A particular type of destination, urban tourism destination, was targeted in this study, mainly due to the variety and abundance of perceived elements.

*3.1. Interview Survey*

3.1.1. Interview Implementation

The interview survey was conducted through face-to-face interviews by four trained graduate students. The interview sites were located in the main tourist sites of Nanjing, a very famous urban tourism destination in China. A stratified random sampling method was used. Sixty respondents were recruited and interviewed in this survey. They ranged in age from 18 to 65 and included slightly more females than males. In the survey, respondents were induced to think of at least two cities they were quite impressed with or had most recently been to in the past four months. The following interview focused on two key questions for each city, and was conducted in a recursive way. The first question was: How do you feel about this city? Respondents were asked to use the one word that first came into their minds to describe it. When the word was given, respondents were asked the following question: *Why do you have this feeling?* They were required to answer in a brief way. This question would be asked round and round until the answer that involved the most concrete elements of the city was given.

3.1.2. Interview Results

There were 37 cities mentioned by the respondents, including both domestic and overseas cities. Domestic cities included Nanjing, Chongqing, Shanghai, Suzhou, Beijing, Xi'an, Guiyang, Yangzhou, Lanzhou, Kunming, Wuhan, Urumqi and Lhasa, while overseas cities included Melbourne, London, Tokyo, Seoul, Gottingen, New York and Paris. A total of 119 descriptive adjectives and 110 named entities referring to cities were collected from the answers (Table 1). Specifically, the answers in different stages were specified as follows.

**Table 1.** Results of interview survey referring to urban destinations.

| Adjectives Traits (N = 119) | Named Entities (N = 110) |
| --- | --- |
| active, amusing, agreeable, attractive*, appealing, attractive, amazing*, alive, beautiful*, brilliant, busy*, charming, calm*, clear, creative*, confident, comfortable*, conventional, classical*, cultural*, compatible*, diligent*, dynamic, depressing, diverse, dull, delicate, energetic, emotional, efficient*, easygoing, elegant*, egocentric, enthusiastic, fancy, fashionable*, funny, fantastic, feminine, friendly, fresh, gentle, generous', good-taste, glamorous*, gorgeous, good-looking, graceful, honest, hard-working, holy, hospitable, idealist, incredible, international, intelligent*, imaginative, important, kind, knowledgeable*, leader, lovely, masculine*, magnificent, mature, mysterious, mannered, natural, native*, neat, original, organized, old*, open-minded*, optimistic, passionate, polite, positive, peaceful, pleasant, promising*, profound, potential, popular, pretty, quiet, relaxed, romantic*, restless, reliable*, religious, responsible, rich*, scholarly, stylish, simple*, stubborn, sincere*, spiritual, showy*, sensible*, sentimental, solemn, successful*, secure, sociable*, traditional, tough*, tolerant, tidy, technical, trendy, unique, up-to-date, vibrant*, virtuous, warm*, welcoming, young | alley, activity, animal, airport, avenue, architecture, bridge*, bar, broadcast media, building, bus, book, beach, celebrity*, cafe shop, campus*, cinema, CBD*, city park*, commodity, communication, city wall*, city hall, driver, dweller, event*, express delivery, food*, folk art, forest, flower*, fast food, festival*, film, famous people*, gate, garden, greenway, gallery, highway, handicrafts*, hotel*, library*, literature*, lake, legend*, museum*, memorial*, mountain*, music, mausoleum*, media*, novel, plant*, policeman, poet, poem, performance, price*, school, street*, pedestrians*, RBD*, resident*, residential area, river*, restaurant*, shopping mall, squire*, seller, stadium, symbol, sea, snack, sky, skyscraper, sculpture*, store, subway, souvenir, tree*, service*, temperature*, temple*, tourist, theme park*, theater, traffic*, tourist guide, transport station*, temperature*, venue, waiter/waitress*, weather*, water*, zoo* |

Note: The traits are listed in alphabetical order; The traits labeled '*' indicate that they were by above 25% of respondents.

Referring to the first question, more than 70% of answers were with respect to the general feelings or holistic atmosphere of a city, such as comfortable, attractive, interesting, agreeable, amazing, unique, modern, beautiful, charming. As a result, 35 adjectives that were mentioned by over 25% of respondents were collected, and are presented in Table 1. Among them, the two most-mentioned adjectives were attractive and comfortable, which were mentioned by more than 37% and 35% of respondents, respectively. The rest of the answers could be divided into two parts: some referred to the more general evaluation, simply as good/terrible/not bad, while the others skipped directly into a more concrete scope that is indicated below.

For the respondents who gave the relatively general evaluation, interviewers further inquired as to why they had these feelings. Referring to this question, more concrete expressions were given, involving numerous and varied aspects of a city, with both tangible and intangible forms, such as buildings, residents, food, weather, folk arts, etc. As a result, 43 nouns that were mentioned by over 25% of the respondents were collected and are shown in Table 1.

### 3.1.3. Hierarchical Fusion Process Hypothesis

Based on both the interview survey and literature review, we originally proposed that there are at least four certain layers in destination image formation, illustrated from bottom to top as follows (Figure 2).

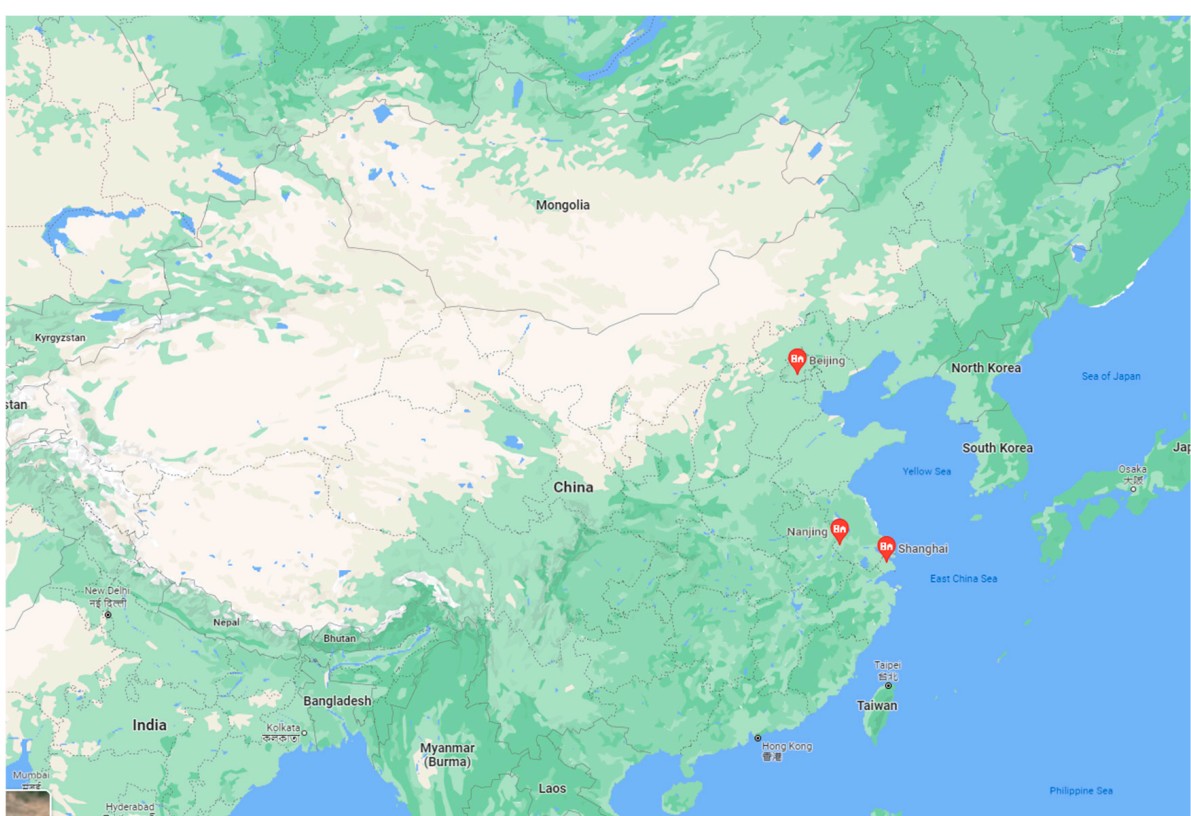

**Figure 2.** The location of Nanjing relative to Beijing and Shanghai on the map of China.

Tourists perceive a destination from various individual elements at first. These elements include both concrete entities that can be directly noticed, such as mountains or buildings, as well as abstract characteristics that are invisible, such as legends or customs [30]. All of these can be viewed as the landscapes of a city. We set it as the first layer of a destination image, and labeled it as the Individual-Landscape Layer (hereinafter referred to as $L_i$).

With the accumulation of landscapes in mind, tourists perceive a destination through fewer abstract characteristics, compared to the elements in the first layer. These characteristics represent the general atmosphere of a city in different aspects. We set this as the second layer of the destination image, and labeled it as the Compound-Atmosphere Layer (hereinafter referred to as $L_c$).

From the interview survey, we found that there were two qualities referring to an urban destination that were most commonly mentioned, i.e., comfortableness and attractiveness. 'Comfortableness' refers to the quality that makes tourists feel relaxed and contented, not only physically but also mentally, while 'attractiveness' refers to the quality that draws tourists' attention to elements that to them feel pleasant and enjoyable. Psychologists Herzberg, Mausner and Snyderman indicate that there are two kinds of factors that influence people's attitudes from positive to negative [31]. One is the hygiene factor, which helps to eliminate people's negative attitudes, but is of no help in the generation of positive attitudes; another is the motivator factor, which contributes to people's positive attitudes, but does not lead to negative attitudes even if it is missing. This idea, breaking away from the traditional views that treat influential effects simply as positive or negative, considers spaces between these two end-points. This helps to understand influential factors in a more elaborate and targeted way. Based on that perspective, we labeled this layer as the Dual-Factor Layer (hereinafter referred to as $L_d$), in an effort to examine whether these two factors constitute a perceived level higher than $L_c$ and can be divided into hygiene and motivator factors, respectively.

Additionally, the overall image, as the most general and abstract characteristic of a tourism destination, is included. This is the top layer, or, in other words, the end point of the destination image formation process, with the label of Overall-Image Layer (hereinafter referred to as $L_o$). Note that, for standardized description, in the following section, we label the perceived elements in two forms, i.e., features and factors. Features can be viewed as the smallest unit/component of a destination image, while factors are the underlying dimensions of features in each layer.

### 3.2. Questionnaire Survey

#### 3.2.1. Questionnaire Design

We prepared a large-scale questionnaire survey, targeting the city of Nanjing. With profound historical cultures, rich literary heritages, and modern urban landscapes, this city attracts millions of visitors from all over the world every year. A field survey was conducted. All of the respondents were tourists who were in the midst of their visits to this city. They were asked to evaluate the city through their direct or indirect contact, using Likert's five-point scale. There were four main parts in the questionnaire that are detailed as follows.

The first part referred to the estimation of elements in $L_i$, the bottom layer of the destination image proposed in this study. Both the interview survey and literature review were used to collect items in this part. After eliminating duplicated items, a total of 44 items were finally included. Respondents were asked to rate each item according to its performance in characterizing and discriminating the city, through Likert's five-point scale, ranging from 1 (no performance) to 5 (excellent performance).

The second part referred to the estimation of elements in $L_c$, a higher level of $L_i$. Based on the theory of anthropomorphism, human-like traits have been accepted as an effective representation of general characteristics concerning a particular object [32]. We asked participants to think of the city as a person, and to estimate the matching rate of that image with a pool of descriptive adjectives, ranging from 1 (not matching at all) to 5 (greatly matching). The items were collected in two ways, i.e., with the urban destination personality scale and the interview survey used to supplement for a specific case. After eliminating the duplicated items, a total of 51 items were finally collected.

The third part referred to the estimation of elements in $L_d$, a higher level of $L_c$, with two items, comfortableness and attractiveness. The same measurement scale referring to

$L_c$ was used here. The existence of this level, as well as whether these two items could be ascribed to hygiene and motivator factors at this level, were explored.

The fourth part referred to the estimation of elements in $L_0$, the highest level of the destination image. In this part, the overall destination image was estimated, ranging from 1 (extremely negative) to 5 (extremely positive), with an induced question: What's your feeling of the city as a whole? Additionally, participants' demographic information, including age, gender, education, profession, income, etc., were also investigated.

### 3.2.2. Questionnaire Collection

The questionnaire survey was carried out in October 2019, which is the most favorite month of the year for the majority of tourists, due to the agreeable weather and national golden week holidays in China. The questionnaires were distributed to the participants at several locations, including airports, railway stations, and famous tourism sites (e.g., Sun Yat-sen's Mausoleum, Confucius Temple, Presidential Palace, Xuanwu Lake Park), using a convenience sampling method. Six trained graduate students were recruited to carry out the survey. A small souvenir was given to each participant as an expression of the investigators' gratitude.

In the end, a total of 800 questionnaires were distributed. After eliminating returned questionnaires of poor quality, 671 completed questionnaires were deemed valid, yielding a valid response rate of 83.9%. Most were from respondents younger than 45 (72.3%), with more from females (53.2%) than males. People who held either a bachelor's degree or above (46%) accounted for the majority. For most of the participants, it was their first or second time (82.8%) to visit this destination. The detailed descriptions of the respondents are shown in Table 2.

**Table 2.** Descriptions of the respondents in questionnaire survey (N = 671).

| Variables | Samples (%) | | Variables | Samples (%) | |
|---|---|---|---|---|---|
| Sex | | | Age | | |
| Male | 314 | 46.8 | 20 and below | 50 | 7.4 |
| Female | 357 | 53.2 | 21 to 34 | 223 | 33.2 |
| Profession | | | 35 to 44 | 151 | 22.5 |
| Government staff | 36 | 5.3 | 45 to 60 | 151 | 22.5 |
| Manager | 55 | 8.2 | 60 or above | 35 | 5.2 |
| Professional/Technical personnel | 45 | 6.6 | Education | | |
| Businessman | 29 | 4.3 | Primary school and below | 29 | 4.3 |
| Staff/Worker | 108 | 16.1 | Middle school | 64 | 9.5 |
| Servicer/Salesman | 115 | 17.2 | High school | 143 | 21.3 |
| Farmer | 15 | 2.3 | Technical college | 127 | 18.9 |
| Student | 193 | 28.7 | Undergraduate | 272 | 40.6 |
| Retired | 52 | 7.8 | Graduate | 36 | 5.4 |
| Other | 23 | 3.5 | Monthly Income | | |
| Visiting Time | | | ¥3000 or below | 171 | 25.5 |
| First time | 317 | 47.3 | ¥3001 to ¥5000 | 313 | 46.6 |
| Second time | 238 | 35.5 | ¥5001 to ¥9999 | 101 | 15.1 |
| Third lime or above | 116 | 17.2 | ¥10,000 or above | 86 | 12.8 |

### 3.3. Data Analysis

Machine learning was applied for data analysis in this study. As the core of artificial intelligence, machine learning discovers potentially useful laws or patterns hidden beneath the data, and constructs models that may gracefully approach the ground-truth relationships in the real world [33,34]. Three machine learning techniques were mainly used in this study, with the implementation of WEKA 3.8, a well-known machine-learning tool box.

Specifically, feature selection was performed as a data pre-processing technique, eliminating the irrelevant and redundant features from the original feature set, and to select the valuable features for the target city; next, feature extraction was applied to extract the main factors, i.e., underlying dimensions, of features in each layer; then, feature ranking was used to extract the influential effects that factors in one layer had on their target concepts in a higher layer.

## 4. Results

We explored the hierarchical fusion process of destination image formation through four proposed layers. Because the perceived elements in higher layers were set as target concepts of the lower layers, and thus needed to be identified first, we explored the four layers from top to bottom. Specifically, every two layers were analyzed in the same phase, which divided the whole process into three phases. In each phase, the effects from the lower layer to higher layer were explored through interactive elements.

### 4.1. Phase 1: Effects from $L_d$ to $L_o$

In this part, we sought to verify and explore the effects from $L_d$ to $L_o$, and to further investigate whether these effects could be divided into hygiene and motivator parts. Since the hygiene (or motivator) factor discriminates the target concept from negative to non-negative (or positive to non-positive) parts, the analysis process in this part was as follows. Referring to the hygiene factor, we set the scores 1 and 2 as negative and 3, 4 and 5 as non-negative in Likert's five-point scale, and then explored the discrimination of comfortableness and attractiveness on overall image through the feature ranking method. Referring to the motivator factor, we set the scores 4 and 5 as positive and 1, 2 and 3 as non-positive, and then used the same method to investigate the discrimination of these two factors on overall image.

#### 4.1.1. Feature Ranking of $L_d$

Feature ranking usually ranks features based on their ability to discriminate a target concept [35]. In this study, comfortableness and attractiveness were set as the factors/features (labeled as $F_{cf}$ and $F_{ar}$, respectively), and overall image was the target concept (labeled as $T_o$). Since information gain (IG) is widely used to characterize the amount of information that the features may contain for discriminating the target concept, we employed information gain values here to rank the discrimination importance of the features to their target concepts.

Taking $F_{cf}$ as an example, information gain values describe the information of $F_{cf}$ 'gained' by splitting the data set according to its different values. This can be formulated as:

$$IG(S, C) \equiv Entropy(S) - \sum_{v \in V_c} \frac{|S_v|}{|S|} Entropy(S_v) \tag{1}$$

where $V_{cf}$ is a set of possible values that $F_{cf}$ can take, and $S_v$ is a subset of $S$, in which $F_{cf}$ takes the value $v$ for all instances, i.e., $S_v = \{s \in S | A(s) = v\}$. Information gain is actually the reduction of entropy if the data set is split using $F_{cf}$. The more reduction, the less the entropy of the resulting subset, and hence the purer in terms of the concept within each subset. That is, $F_{cf}$ helps more in distinguishing between concepts.

In this study, the gain threshold was set as 0.4, according to Schaffer, Whitley and Eshelman [36]. This indicated that the discrimination importance of the features that either reached or exceeded 0.4 could be considered to have great importance for $T_o$. The higher the value was, the greater the discrimination importance could be. There would be no importance when the value reached 0.

The results indicate that both $F_{cf}$ ($IG_{cf\_h} = 0.853$) and $F_{ar}$ ($IG_{ar\_m} = 0.772$) have discrimination importance to $T_o$. Based on that, the existence of $L_d$, as well as the effects from $L_d$ to $L_o$, were verified. This indicated that overall destination image is formed through the

interaction of two characteristics, i.e., comfortableness and attractiveness, of a destination, from tourists' perceptions.

### 4.1.2. Identifying Hygiene vs. Motivator Factors

Based on the values of information gain, $F_{cf}$ has great importance in discriminating $T_o$ from negative to non-negative ($IG_{cf\_h} = 0.853$), rather than from positive to non-positive ($IG_{cf\_m} = 0.321$), and therefore can be treated as hygiene factor of $T_{o\_h}$. This indicates that tourists who feel that the destination is uncomfortable, no matter physiologically or psychologically, would have negative perceptions of it, whereas if they feel that the destination is comfortable, it does not mean that they would have positive perceptions. On the contrary, $F_{ar}$ has great importance in discriminating $T_o$ from positive to non-positive ($IG_{ar\_m} = 0.772$), rather than from negative to non-negative ($IG_{ar\_h} = 0.259$), and therefore can be treated as motivator factor of $T_{o\_m}$. This means that tourists who feel that the destination is attractive would have positive perceptions of it, whereas if they feel that the destination is unattractive, it does not mean that they would have negative perceptions.

### 4.2. Phase 2: Effects from $L_c$ to $L_d$

With the identification of $L_d$, the effects from $L_c$ to $L_d$ were explored next. First, the appropriate features in $L_c$ were selected through an original pool of items that made the final included ones tailored for the target city. Second, the main factors (i.e., underlying dimensions) of $L_c$ were extracted, and the effects of the factors from $L_c$ to $L_d$ were further explored.

### 4.2.1. Feature Selection of $L_c$

Feature selection, as a data-processing technique, aims to select features with all the useful information sufficient for learning the target concept. To this end, the irrelevant and redundant features are eliminated from the original feature set. One widely used feature selection technique, wrapper-based feature selection, was applied in this study.

Specifically, wrapper-based feature selection was used to select a pool of features that were most relevant to their target concepts in $L_d$, i.e., comfortableness and attractiveness. The features that were relevant to either of these two target concepts were retained in the final feature set.

If we denote the original feature set as $F_c = \{F_{c1}, F_{c2}, \dots F_{c44}, \}$, a set of selected informative features $F_c^*$ can be obtained by solving the following optimization problem as:

$$F_c^* = arg \min_{F_c' \subset F_c} \mathcal{L}\left(F_c', \left\{T_{cf}, T_{ar}\right\}\right) \tag{2}$$

where $F_c$ is the original feature set, $T_{cf}$ and $T_{ar}$ are the target concepts corresponding to comfortableness and attractiveness, respectively, and $\mathcal{L}\left(F_c', T\right)$ is the loss for predicting the target concept using the feature subset $F_c'$.

A classic zero-order randomized optimization approach, genetic algorithm (GA), was employed in this study to solve the optimization problem in Equation (2) and identify a proper feature subset $F_c^*$, such that the loss or errors for predicting the target concept using the feature subset were minimized.

According to the notions of natural evolution, the genetic algorithm was introduced to create artificial systems that work on a similar basis. In this study, the size of population $N$ was set to 100, the probability of mutation $P_m$ was set to 0.005, the probability of cross-over $P_c$ was set to 0.6, and the maximum number of evolution rounds $T$ was set to 500. In each round of evolution, a roulette wheel selection was used to randomly select individuals for cross-over based on $P_c$, and then to select individuals for mutation based on $P_m$. Afterwards, the fitness score $f$ was computed for each individual. The higher the fitness score, the more likely that this individual could survive to the next generation. Such evolution iterates either until convergence or the maximum number of iterations is reached. The final solution was obtained with the highest fitness $f_c = 0.0697$ among the populations.

Table 3 shows the optional feature subset $F_c^{**} = \{F_{c1}^* \cup F_{c2}^*\}$ that has great influence on either comfortableness or attractiveness of a destination. Twenty-nine traits were retained in this final feature set, whereas other traits, such as aggressive, courteous, excited, etc., were eliminated from the original feature set, due to their ineffectiveness in discriminating target concepts and therefore were not involved in the subsequent studies.

**Table 3.** Feature selection results of perceived elements in $L_c$.

| Features | $T_{cf}$ | $T_{ar}$ | Features | $T_{cf}$ | $T_{ar}$ | Features | $T_{cf}$ | $T_{ar}$ | Features | $T_{cf}$ | $T_{ar}$ |
|---|---|---|---|---|---|---|---|---|---|---|---|
| Aggressive | | | Diligent | √ | | Knowledgeable | | | Self-confident | | |
| Artistic | | √ | Down-to-earth | | | Masculine | | √ | Simple | √ | |
| Beautiful | √ | √ | Efficient | √ | | Mysterious | | | Sociable | | |
| Busy | √ | | Excited | | | Native | | √ | Sincere | √ | |
| Courteous | | | Elegant | √ | √ | Old | | √ | Showy | | |
| Calm | √ | √ | Fashionable | √ | √ | Open-minded | √ | | Sensible | | |
| Classical | | √ | Glamorous | | √ | Outdoorsy | | | Successful | | |
| Cultural | √ | √ | Generous | √ | √ | Promising | | √ | Tough | √ | |
| Cheerful | | √ | Home-oriented | | | Rich | | | Tolerant | √ | |
| Compatible | √ | | Intelligent | √ | | Romantic | | √ | Vibrant | | √ |
| Creative | | √ | Independent | | | Reliable | √ | | Warm | √ | √ |
| | | | | | Highest Fitness: Y = 0.0697 | | | | | | |

The items with no '√' labeled on the right side are excluded from the final pool; $T_{cf}$ and $T_{ar}$ are the target concepts corresponding to the four factors in $L_c$.

### 4.2.2. Feature Extraction of $L_c$

Feature extraction is used for extracting the latent structure from data. Principal component analysis (PCA), a powerful unsupervised feature extraction method, was employed in this study. The goal of PCA is to reduce the dimensionality of data while retaining as much of the spreading tendency of the original dataset as possible. The identified spreading directions of the original data, which are usually a linear combination of the original features, are regarded as the latent structure of the data.

PCA-based feature extraction was employed here to identify underlying factors of $L_c$. Four main factors were finally identified, accounting for 78.6% of the total variance (Table 4). The first factor included features that mainly represented local cultural characteristics of a city that make it quite different from other cities, and therefore was labeled as *uniqueness* ($F_{uq}$). The second factor included features that represented the warmth and acceptance that a city shows to its visitors, and thus was labeled as kindness ($F_{kd}$). The third factor included features that described the liveliness and openness of a city, and therefore was named as activeness ($F_{ai}$). The features in the fourth factor referred to the actual and potential development strength that a city has, and thus this factor was named as competence ($F_{cp}$). The features included in these four main factors are presented in Table 4.

**Table 4.** PCA results of perceived elements in $L_c$.

| Factors/Features | Factor Loading | | | | Eigenvalue | Cumulative Explained Variance |
|---|---|---|---|---|---|---|
| | $F_{uq}$ | $F_{kd}$ | $F_{ai}$ | $F_{cp}$ | | |
| $F_{uq}$: Uniqueness | | | | | 8.31 | 26.4% |
| $F_{uq\_1}$: Classical | 0.832 | 0.272 | 0.216 | 0.121 | | |
| $F_{uq\_2}$: Cultural | 0.807 | 0.325 | 0.191 | 0.031 | | |
| $F_{uq\_3}$: Native | 0.747 | 0.218 | 0.063 | 0.297 | | |
| $F_{uq\_4}$: Glamorous | 0.623 | 0.403 | 0.223 | 0.213 | | |
| $F_{uq\_5}$: Beautiful | 0.533 | 0.123 | 0.071 | 0.037 | | |
| $F_{uq\_6}$: Artistic | 0.509 | 0.280 | 0.040 | 0.283 | | |
| $F_{uq\_7}$: Elegant | 0.431 | 0.131 | 0.193 | 0.162 | | |
| $F_{uq\_8}$: Romantic | 0.402 | 0.052 | 0.040 | 0.073 | | |

**Table 4.** *Cont.*

| Factors/Features | Factor Loading | | | | Eigenvalue | Cumulative Explained Variance |
|---|---|---|---|---|---|---|
| | $F_{uq}$ | $F_{kd}$ | $F_{ai}$ | $F_{cp}$ | | |
| $F_{kd}$: Kindness | | | | | 7.22 | 49.3% |
| $F_{kd\_1}$: Warm | 0.320 | 0.819 | 0.206 | 0.216 | | |
| $F_{kd\_2}$: Simple | 0.287 | 0.783 | 0.305 | 0.200 | | |
| $F_{kd\_3}$: Generous | 0.039 | 0.701 | 0.127 | 0.313 | | |
| $F_{kd\_4}$: Compatible | 0.035 | 0.687 | 0.388 | 0.273 | | |
| $F_{kd\_5}$: Tolerant | 0.082 | 0.635 | 0.193 | 0.071 | | |
| $F_{kd\_6}$: Reliable | 0.289 | 0.577 | 0.390 | 0.129 | | |
| $F_{kd\_7}$: Sincere | 0.271 | 0.516 | 0.283 | 0.206 | | |
| $F_{ai}$: Activeness | | | | | 6.40 | 65.2% |
| $F_{ai\_1}$: Old | 0.105 | 0.367 | 0.818 | 0.201 | | |
| $F_{ai\_2}$: Masculine | 0.308 | 0.136 | 0.760 | 0.397 | | |
| $F_{ai\_3}$: Fashionable | 0.299 | 0.280 | 0.729 | 0.057 | | |
| $F_{ai\_4}$: Open-minded | 0.170 | 0.223 | 0.646 | 0.309 | | |
| $F_{ai\_5}$: Calm | 0.283 | 0.077 | 0.529 | 0.133 | | |
| $F_{ai\_6}$: Cheerful | 0.019 | 0.103 | 0.437 | 0.237 | | |
| $F_{ai\_7}$: Vibrant | 0.249 | 0.276 | 0.408 | 0.006 | | |
| $F_{cp}$: Competence | | | | | 5.01 | 78.6% |
| $F_{cp\_1}$: Intelligent | 0.234 | 0.293 | 0.134 | 0.809 | | |
| $F_{cp\_2}$: Tough | 0.187 | 0.201 | 0.059 | 0.753 | | |
| $F_{cp\_3}$: Promising | 0.237 | 0.234 | 0.032 | 0.667 | | |
| $F_{cp\_4}$: Creative | 0.107 | 0.070 | 0.280 | 0.621 | | |
| $F_{cp\_5}$: Diligent | 0.255 | 0.135 | 0.196 | 0.552 | | |
| $F_{cp\_6}$: Efficient | 0.208 | 0.193 | 0.200 | 0.478 | | |
| $F_{cp\_7}$: Busy | 0.310 | 0.079 | 0.156 | 0.434 | | |

### 4.2.3. Feature Ranking of $L_c$

Feature ranking was used to explore the effects from $L_c$ to $L_d$, i.e., from the factors of uniqueness, kindness, activeness and competence, to their target concepts of comfortableness and attractiveness. Information gain was also used here to rank the discrimination importance of the factors, with the gain threshold being set as 0.4.

Specifically, referring to $T_{cf}$, there were three factors that were important in discriminating it, i.e., kindness ($IG_{kd\_cf} = 0.836$), competence ($IG_{cp\_cf} = 0.721$), and activeness ($IG_{ai\_cf} = 0.669$), ranking in descending order. Among them, *kindness* had the greatest importance, which indicates that psychological comfort is what tourists care more about, compared with physiological comfortableness. Uniqueness ($IG_{uq\_cf} = 0.306$) had no importance in discrimination.

Referring to $T_{ar}$, there were also three factors that were important in discriminating it, i.e., uniqueness ($IG_{uq\_ar} = 0.892$), kindness ($IG_{kd\_ar} = 0.737$), and activeness ($IG_{ai\_ar} = 0.609$), ranking in descending order. Among them, uniqueness had the greatest importance, which confirmed its crucial role in attracting tourists. Competence ($IG_{cp\_ar} = 0.219$) had no importance in discrimination.

Note that, referring to all four of these factors, kindness and activeness were important in discriminating both of the two target concepts, while uniqueness and competence could discriminate only one of them. From the results mentioned above, the existence of $L_c$ as well as the effects from $L_c$ to $L_d$ were verified. This indicates that, from tourists' perceptions, the comfortableness and attractiveness of a destination are formed through four types of atmosphere, i.e., uniqueness, kindness, activeness and competence, which interact with each other in different ways.

### 4.3. Phase 3: Effects from $L_i$ to $L_c$

With the identification of $L_c$, the effects from $L_i$ to $L_c$ were explored next. First, the appropriate features in $L_i$ were selected through an original pool of items that made the

final included ones tailored for the target city. Second, the main factors of $L_i$ were extracted, and the effects of the factors, from $L_i$ to $L_c$, were further explored.

### 4.3.1. Feature Selection of $L_i$

Feature selection was used to select feature subset $F_i^*$, which would be most relevant to influence tourists' perceptions of $L_c$ from the original feature set $F_i = \{F_{i1}, F_{i2}, \ldots, F_{i51}\}$. Here, $F_i^*$ could be obtained by solving the following optimization problem as:

$$F_i^* = \underset{F_i' F_i}{arg\min} F\left(F_i', \{T_{uq}, T_{kd}, T_{ai}, T_{cp}\}\right) \tag{3}$$

where $T_{uq}$, $T_{kd}$, $T_{ai}$ and $T_{cp}$ are the target concepts corresponding to four factors of $L_c$, i.e., uniqueness, kindness, activeness and competence, respectively. The parameters, i.e., the size of population $N$, the probability of mutation $P_m$, the probability of cross-over $P_c$, and the maximum number of evolution rounds $T$, were set the same as those in feature selection of $L_c$. In the end of the last iteration, the final solution was obtained in the form of 35 features with the highest fitness $f = 0.0732$ among the populations.

Table 5 shows the optional feature subset $F_i^{**} = \{F_{i1}^* \cup F_{i2}^* \cup F_{i3}^* \cup F_{i4}^*\}$ that has great influence on $L_c$. Thirty-five features that could discriminate any of the four target concepts were included in the final pool, whereas other items, such as theme park, mountain scenery, municipal building, etc., were eliminated from the original list and therefore were not involved in the subsequent studies, due to their ineffectiveness in discriminating the target concepts.

**Table 5.** Feature selection results of perceived elements in $L_i$.

| Features | $T_{uq}$ | $T_{kd}$ | $T_{ai}$ | $T_{cp}$ | Features | $T_{uq}$ | $T_{kd}$ | $T_{ai}$ | $T_{cp}$ |
|---|---|---|---|---|---|---|---|---|---|
| Avenue/Alley | √ | | √ | | Mountain scenery | | | | |
| Animal | | | | | Municipal building | | | | |
| Architectural style | | | | | Plant | √ | | | √ |
| Bridge | | √ | √ | | Price level | | √ | | √ |
| Bar street | | | | | Public historical event | | | √ | |
| City wall | √ | | √ | | Pedestrian street | √ | | √ | √ |
| Campus | √ | √ | √ | | Public service | | √ | | √ |
| City squire/park | √ | | √ | √ | Resident | | √ | √ | √ |
| CBD& RBD | | | √ | √ | Restaurant and hotel | √ | √ | √ | √ |
| Cultural venue | √ | | √ | √ | Residential area | | | | |
| Commodity and souvenir | √ | | | √ | Skyscraper | | | √ | √ |
| Communication | | | | √ | Stadium | | | | |
| Express delivery | | | | | Street sculpture | | √ | √ | |
| Famous building | √ | | √ | √ | Sanitary | | √ | | √ |
| Film and television work | | | | | Security | | √ | | √ |
| Festival and celebration | | | √ | √ | Shopping mall | | | | |
| Folk art and handicraft | √ | | √ | | Snack Shop | √ | √ | | |
| Famous public people | | | √ | √ | Temple/tower/mausoleum | √ | √ | √ | |
| Historical garden | | | | | Theme park/Zoo | | | | |
| Industrial area | | | | | Traffic | | | √ | √ |
| Local legend/story/music | | | √ | √ | Tourism related service | | √ | | √ |
| Local Food | √ | √ | | | Transport station | | √ | √ | √ |
| Literature work | √ | √ | √ | | Traditional street and area | √ | | √ | |
| Local media | | | | | Water (front) scenery | √ | √ | √ | |
| Museum/Memorial | √ | | √ | | Weather and temperature | | | | |
| Main road | | | | | | | | | |

Highest Fitness: Y = 0.0732

The items with no '√' labeled on the right side are excluded from the final pool; $T_{uq}$, $T_{kd}$, $T_{ai}$ and $T_{cp}$ are the target concepts corresponding to the four factors in $L_c$.

### 4.3.2. Feature Extraction of $L_i$

PCA was implemented to identify the underlying factors of landscape elements in $L_i$. Five main factors were finally identified, accounting for 79.5% of the total variance (Table 6). The first factor included elements that could be viewed as the most typical representations of the city, with respect to the city's culture, nature, history, economy, etc., and therefore were labeled as landmark ($F_{ld}$). The second factor included elements that could be viewed as the belongings to the old periods, with both tangible and intangible forms, and therefore was labeled as relic ($F_{ri}$). The third factor included elements that could be viewed as physical spaces or entities, representing urban modernization from various aspects, and therefore was labeled as modernity ($F_{me}$). The fourth factor included elements that were related to living style or environment of a city that tourists may come into direct contact with, and therefore was labeled as living ($F_{li}$). The fifth factor included elements that could be viewed as public facilities or services that a city offers to its visitors, and therefore was labeled as service ($F_{sv}$). The specific features in each factor are presented in Table 6.

**Table 6.** PCA results for perceived elements in $L_i$.

| Factors/Features | Factor Loading | | | | | Eigenvalue | Cumulative Explained Variance |
|---|---|---|---|---|---|---|---|
| | $F_{ld}$ | $F_{ri}$ | $F_{me}$ | $F_{li}$ | $F_{sv}$ | | |
| $F_{ld}$: Landmark | | | | | | 9.54 | 27.9% |
| $F_{ld\_1}$: Temple/tower/mausoleum | | | 0.294 | 0.135 | 0.223 | | |
| $F_{ld\_2}$: City wall | | | 0.137 | 0.332 | 0.003 | | |
| $F_{ld\_3}$: Bridge | | | 0.302 | 0.141 | 0.172 | | |
| $F_{ld\_4}$: Famous building | | | 0.234 | 0.367 | 0.312 | | |
| $F_{ld\_5}$: Museum and memorial | | | 0.085 | 0.285 | 0.183 | | |
| $F_{ld\_6}$: Public historical event | | | 0.093 | 0.123 | 0.192 | | |
| $F_{ld\_7}$: Avenue/alley | | | 0.075 | 0.292 | 0.234 | | |
| $F_{ld\_8}$: Street sculpture | | | 0.188 | 0.233 | 0.008 | | |
| $F_{ri}$: Relic | | | | | | 8.39 | 47.1% |
| $F_{ri\_1}$: Traditional street and area | | | 0.123 | 0.300 | 0.083 | | |
| $F_{ri\_2}$: Famous public people | | | 0.273 | 0.256 | 0.193 | | |
| $F_{ri\_3}$: Campus | | | 0.028 | 0.283 | 0.128 | | |
| $F_{ri\_4}$: Folk art and handicraft | | | 0.311 | 0.276 | 0.103 | | |
| $F_{ri\_5}$: Literature work | | | 0.192 | 0.321 | 0.135 | | |
| $F_{ri\_6}$: Local legend/story/music | | | 0.225 | 0.006 | 0.106 | | |
| $F_{ri\_7}$: Water (front) scenery | | | 0.302 | 0.175 | 0.272 | | |
| $F_{me}$: Modernity | | | | | | 6.27 | 62.8% |
| $F_{me\_1}$: CBD&RBD | | | 0.713 | 0.103 | 0.209 | | |
| $F_{me\_2}$: Transport station | | | 0.629 | 0.115 | 0.004 | | |
| $F_{me\_3}$: Restaurant and hotel | | | 0.600 | 0.304 | 0.183 | | |
| $F_{me\_4}$: Skyscraper | | | 0.573 | 0.231 | 0.222 | | |
| $F_{me\_5}$: Commodity and souvenir | | | 0.551 | 0.093 | 0.179 | | |
| $F_{me\_6}$: Pedestrian street | | | 0.523 | 0.173 | 0.271 | | |
| $F_{me\_7}$: City park and squire | | | 0.432 | 0.055 | 0.338 | | |
| $F_{li}$: Living | | | | | | 5.53 | 71.9% |
| $F_{li\_1}$: Resident | | | 0.191 | 0.747 | 0.091 | | |
| $F_{li\_2}$: Traffic | | | 0.258 | 0.639 | 0.033 | | |
| $F_{li\_3}$: Local food | | | 0.199 | 0.561 | 0.382 | | |
| $F_{li\_4}$: Cultural venue | | | 0.205 | 0.532 | 0.245 | | |
| $F_{li\_5}$: Festival and celebration | | | 0.316 | 0.513 | 0.234 | | |
| $F_{li\_6}$: Snack shop | | | 0.352 | 0.477 | 0.178 | | |
| $F_{li\_7}$: Plant | | | 0.067 | 0.420 | 0.101 | | |
| $F_{sv}$: Service | | | | | | 4.90 | 79.5% |
| $F_{sv\_1}$: Tourism related service | | | 0.134 | 0.008 | 0.802 | | |
| $F_{sv\_2}$: Sanitary | | | 0.372 | 0.171 | 0.633 | | |
| $F_{sv\_3}$: Public service | | | 0.251 | 0.041 | 0.610 | | |
| $F_{sv\_4}$: Security | | | 0.299 | 0.124 | 0.581 | | |
| $F_{sv\_5}$: Price level | | | 0.005 | 0.233 | 0.502 | | |
| $F_{sv\_6}$: Communication | | | 0.348 | 0.376 | 0.433 | | |

### 4.3.3. Feature Ranking of $L_i$

Feature ranking was used to explore the effects from $L_i$ to $L_c$, i.e., from the factors of landmark, relic, modernity, living, and service, to their target concepts, i.e., kindness, uniqueness, activeness, and competence. Information gain was employed here, with the gain threshold being set as 0.4.

Specifically, referring to the target concept of *uniqueness*, four factors had great importance in distinguishing it, i.e., landmark ($IG_{ld\_qu}$ = 0.852), relic ($IG_{ri\_qu}$ = 0.771), modernity ($IG_{me\_qu}$ = 0.503), and living ($IG_{li\_qu}$ = 0.435), ranked in descending order. Landmark had the most important effect, mainly due to landmarks' greatest symbol significance of a city as well as highest popularity for tourists. Referring to the target concept of *kindness*, three factors had great importance in distinguishing it, i.e., service ($IG_{sv\_kd}$ = 0.807), living ($IG_{li\_kd}$ = 0.780), and modernity ($IG_{me\_kd}$ = 0.523). *Service* had the greatest effect, mainly due to the attitudes of servicers as well as the quality of their services in greatly reflecting the warmth and acceptance of the city. There were three factors that had great importance in distinguishing the target concept of activeness, i.e., *relic* ($IG_{ri\_ai}$ = 0.792), landmark ($IG_{ld\_ai}$ = 0.658), and modernity ($IG_{me\_ai}$ = 0.453). *Relic* had the greatest effect, mainly due to the age of relics as well as the number of relics distributed in the city, in influencing the city's activeness that tourists may easily perceive. Additionally, there were four factors that had great importance in distinguishing the target concept of *competence*, i.e., modernity ($IG_{me\_cp}$ = 0.788), service ($IG_{sv\_cp}$ = 0.667), landmark ($IG_{ld\_cp}$ = 0.639), and living ($IG_{li\_cp}$ = 0.465). Modernity and service, as both hardware and software conditions of a city, could be viewed as the indexes of urban modernization, and therefore had the most important effects.

From the results mentioned above, the existence of $L_c$ as well as the effects from $L_c$ to $L_d$ were verified. The results indicate that, from tourists' perceptions, the city's atmosphere of uniqueness, kindness, activeness and competence was formed through all five types of landscapes, i.e., landmark, relic, modernity, living, and service, which interact with each other in different ways. Table 7 shows the effects of the elements on their higher-layer targets through information gain values, ranging from phase 1 to phase 3. The effects of elements in the hierarchy of destination image formation are also illustrated in Figure 3.

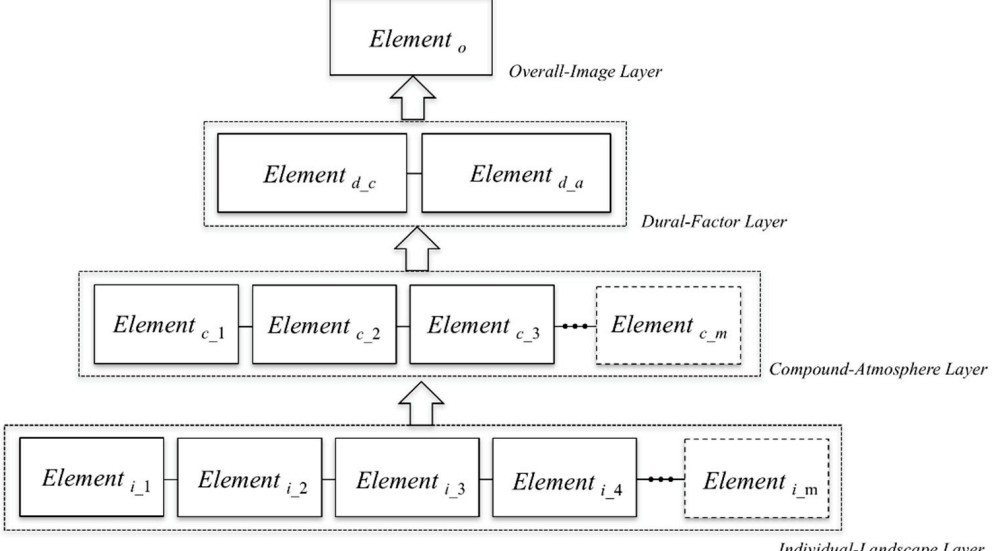

**Figure 3.** Hypothesis of hierarchical fusion process of destination image formation.

**Table 7.** Information gain values of perceived elements to their target concepts in three phases.

| | Phase1 | | | Phase2 | | | Phase3 | | | |
|---|---|---|---|---|---|---|---|---|---|---|
| | $T_{o\_h}$ | $T_{o\_m}$ | | $T_{cf}$ | $T_{ar}$ | | $T_{uq}$ | $T_{kd}$ | $T_{ai}$ | $T_{cp}$ |
| $F_{cf}$ | 0.853 | 0.321 | $F_{uq}$ | 0.306 | 0.892 | $F_{ld}$ | 0.852 | 0.372 | 0.658 | 0.639 |
| $F_{ar}$ | 0.259 | 0.772 | $F_{kd}$ | 0.836 | 0.737 | $F_{ri}$ | 0.771 | 0.236 | 0.792 | 0.380 |
| | | | $F_{ai}$ | 0.669 | 0.601 | $F_{me}$ | 0.503 | 0.523 | 0.453 | 0.788 |
| | | | $F_{cp}$ | 0.721 | 0.219 | $F_{li}$ | 0.435 | 0.780 | 0.227 | 0.465 |
| | | | | | | $F_{sv}$ | 0.321 | 0.807 | 0.313 | 0.667 |

## 5. Discussion

The findings in this study verified that destination image formation is indeed a hierarchical fusion process, including interactive elements in at least four layers. At first, through direct or indirect contact, people acquire an original image of a destination through landscape elements referring to it. These elements are commonly numerous, discrete and concrete and can be viewed as the smallest units of destination image, involved with five types of landscapes, i.e., landmark, relic, modernity, living and service. This is the first layer of a destination image.

With the accumulation of landscape elements that tourists perceive, these elements interact with each other so as to fuse into new ones. The new elements are less in number, but more general and abstract, referring to four types of compound characteristics, such as kindness, uniqueness, activeness, and competence. Specifically, within tourists' perceptions, the interaction of landmark, relic, modernity, and living contributes to generate the atmosphere of uniqueness referring to a particular city; with the interaction of service, living, and modernity, the atmosphere of kindness is generated; relic, landmark, and modernity interact with each other so as to create the atmosphere of activeness; additionally, with the interaction of modernity, service, landmark, and living, the atmosphere of competence is generated. This is the second layer of destination image, with the elements within it formed through the interaction of first-layer elements in different ways. Note that, for a common target, the effects of elements in lower levels are quite different.

Through deeper contact with the destination, the atmospheres in different types are interacted and fused again so as to form into two new ones in tourists' minds, i.e., comfortableness and attractiveness. 'Comfortableness' refers to the quality of a destination that makes tourists feel relaxed and contented. It is generated from the interaction of three characteristics in the lower layer, i.e., kindness, competence and activeness. 'Attractiveness' refers to the quality that draws tourists' attention to it and to them feels pleasant and enjoyable. It is generated from the interaction of three characteristics, i.e., uniqueness, kindness, and activeness. These two factors are more general and abstract, compared with those in the second layer, and constitute the third layer of destination image.

Through the interaction of both comfortableness and attractiveness of a destination that tourists perceive, the most general image is finally formed, i.e., overall image. This is the top layer of destination image formation. These two factors have different effects on the formation of overall image. Comfortableness helps to eliminate tourists' negative perceptions of a city, and therefore can be ascribed to the hygiene factor, while attractiveness contributes to produce positive perceptions of a city, and thus can be ascribed to the motivator factor. This is the first attempt to explore tourism destination image through dual-factor theory, which is commonly used in research on human resource management and consumer behavior [37,38]. Moreover, unlike previous research that divides dual factors into either hygiene or motivator factors, this study found that dual factors can be compound constructs, generated from different combinations of elements.

Generally, destination image formation can be viewed as a dynamic hierarchical process. From bottom to top, the layers of destination image formation are individual-landscape layer, compound-atmosphere layer, dual-factor layer, and overall-image layer, respectively. The perceived elements in different layers are formed from more to less,

separate to compound, and concrete to abstract. The elements in each layer interact with each other so as to fuse into new ones in the higher layer. This process proceeds level by level until the formation of the most general and abstract one, i.e., overall destination image. This is a dynamic fusion process within a pyramid hierarchical structure (Figure 4).

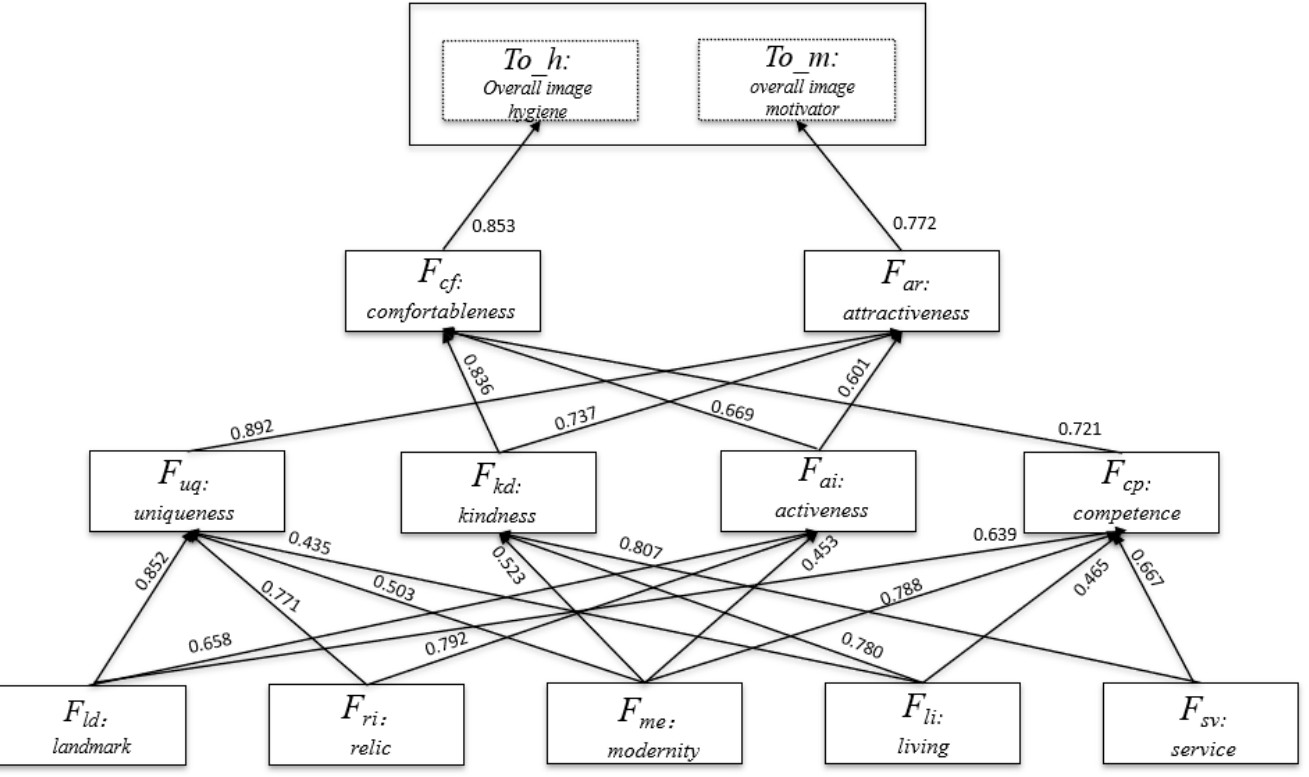

**Figure 4.** Information gain values of perceived elements in different layers.

## 6. Conclusions

Destination image is one of the core concepts in tourism destination research. Scholars accept that destination image consists of perceived elements that have been explored commonly through static structure or external effects (referring to antecedents or consequences). However, the internal effects referring to these elements have been largely ignored. In view of this, this study, thinking outside the box, sought to explore destination image formation through the interactions of perceived elements that would therefore be considered within a dynamic, synthetic system.

Based on both interview and questionnaire surveys, we not only verified the existence of interactive elements, but also found out how these elements interact with each other in the process of destination image formation. An urban tourism destination was selected as the target case in this study. Through machine learning techniques, we finally found that: perceived elements do interact with each other; with the interaction of elements in different ways, new forms of elements are generated in higher levels; this process continues level by level, with the elements being from more to less, separate to compound, and concrete to abstract; and destination image formation can therefore be viewed as a dynamic fusion process, within a pyramid hierarchical structure.

### 6.1. Theoretical Contribution

The theoretical contribution of this study is to explore destination image formation from a new perspective, i.e., focusing on the interactions of perceived elements. As a result, the hierarchical fusion process of destination image formation was first detected and identified in tourism research. Methodologically, the machine learning method, as the

core of artificial intelligence, was applied in this study, providing exemplary results for intelligent data analysis in quantitative tourism research.

### 6.2. Management Significance

The findings provide practical insight into brand image construction of urban tourism destinations in a more comprehensive and targeted way. When building the brand image of urban tourism destinations, managers should not only consider a single element in the process of destination image formation, but should also explore how these elements interact, and this process is systematic and progressive. Brand image building can be regarded as a dynamic integration process within the pyramid hierarchy, with the elements being from more to less, separate to compound, and concrete to abstract.

### 6.3. Limitations and Prospects

This study has limitations as well that would be addressed in our future work. First, the data used in this study were all collected from on-site surveys, while online data (such as data in travel blogs or online social media) could be further utilized for their superiority in terms of volume and objectivity. Second, we selected one particular urban destination as the target case, because the main purpose of this study was to verify the existence of interactive elements as well as their dynamic hierarchy in destination image formation. More cities, as well as more types of tourist destinations, would be targeted in our future work. Furthermore, the variation in travel phases would be further considered in this hierarchical fusion process to identify how perceived elements interact with each other in different travel phases.

**Author Contributions:** Writing—original draft, X.Z.; Writing—review & editing, C.Z.; Resources, Y.L. and Z.X.; Funding acquisition, Z.H. All authors have read and agreed to the published version of the manuscript.

**Funding:** This research was funded by National Natural Science Foundation of China, grant number 41871141, 41671137, 41301145, 42101218; the Tourism Young Expert Training Program, grant numberTYETP201526; and Humanities and Social Science Research Youth Fund Project of Ministry of Education, grant number21YJC790131.

**Institutional Review Board Statement:** Not applicable.

**Informed Consent Statement:** Not applicable.

**Data Availability Statement:** Data sharing not applicable.

**Conflicts of Interest:** The authors declare no conflict of interest.

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
