# Peer review of "Hierarchical Fusion Process of Destination Image Formation: Targeting on Urban Tourism Destination"

_sustainability, doi:10.3390/su132111805_

Round 1

Reviewer 1 Report

The paper has good potential and is a novel way of looking at destination image. However, I would like to point out some points for your consideration to improve the paper.

  • The figures and the tables referred in the paper is not in the paper. Hence, I am unable to follow some of the arguments/discussion in reference to that.
  • The paper needs a thorough check for language to further improve the quality of the paper. For example, some choice of words do not seem to fit or explain what authors intend to.
  • The abstract can be improved. At the moment I find that there is lack of clarity what the authors are trying to explain. There is a sort of vagueness to the whole abstract.
  • Paragraph 2 and 3 in the introduction is the same.
  • In some places, a reference would make your argument stronger. For example, 'Tourism destination image is one of the core concepts in tourism research. This concept is originally derived from psychology field, and has been introduced and developed 19 in multi-disciplines, such as biology, geography, and sociology' (Ref).
  • Unfortunately, at the moment, the conceptual framework is poorly developed. Destination image is a much studied concept and there is a body of literature to review to develop a robust and a clear conceptual framework. Especially, in a quantitative study when you have to rely on the construct(s) to develop the relationship between the different factors, a strong foundation is important. The idea is in the paper - but I think it can be executed better. It may help if you do not rush in the theoretical framework section, instead take time to fully develop the constructs and the framework. This point is particularly jarring given the extent body of rich literature on destination image. 
  • Please do not end sentences with et al. 'Domestic cities refer to Nanjing, Chongqing, Shanghai, Suzhou, Beijing, Xi’an, 160 Guiyang, et al.'. List all the cities, it is a finding from your interviews.
  • How was the interview data analysed? Including such information will improve the quality of the paper, including being transparent about the process undertaken.
  • I am having a difficulty agreeing that the hierarchical process happened organically in your study and that it can be observed from your findings. Perhaps more elaboration about your phase 1 might help in this instance? (Interview survey is used to preliminarily verify the existence of hierarchy in destination image formation, as well as to collect relevant perceived elements as many as possible).
  • An additional idea to consider is: there are equally 3 good papers here. 1] Develop the conceptual framework to argue the existence of the hierarchical process and transformation of the perceived elements (that contribute to forming the DI); 2] A qualitative paper that focus on the interview phase to argue the hierarchical process, 3] the Quantitative survey paper. 2 & 3 can also be combined if you do not want to separate them, although you will get more value to publish it as two separate papers. 
  • I wish you all the best with your publication

Author Response

Thanks for your suggestions, the following is my response :

1.The figures and the tables referred in the paper have been revised in the paper, please check it.

2. I have made relevant changes to the language of the paper.

3.Abstract: Image has been widely accepted as a combination of perceived elements which are commonly discrete and static ones. ‘Discrete’ means that the elements are treated as separate ones with each other, with no interactions among them. ‘Static’ means that the elements would not be changed into other forms in the process of destination image formation. This study, thinking outside the box, tries to explore destination image formation through perceived elements, taking their interactions and corresponding changes into account. Machine learning, as the core of artificial intelligence, is applied for data analysis in this study. Urban tourism destination is targeted because of its variety and abundance of perceived elements. Data are collected from both interview and questionnaire survey of tourists. Through several phases of analysis, this study finally finds that, in tourism destination image formation, perceived elements do interact with each other and change into new forms level by level. Specifically, there are at least four levels in the whole process of destination image formation, i.e., individual-landscape layer, compound-atmosphere layer, dual-factor layer, and overall-image layer, from bottom to top. In the bottom stage, elements are commonly numerous, separate, and concrete. With the interactive effects of the elements, they integrate with each other and generate some fewer new forms in higher level, which would be more general and abstract. This process goes on and on until the formation of the most general one, i.e., overall destination image. Based on the findings, the dynamic fusion process and pyramid hierarchy of destination image formation is disclosed. This study explores destination image formation from a brand-new perspective, considering perceived elements within a dynamic, synthetic system, and therefore provides practical insights into destination image construction in a more comprehensive and targeted way.

4.The same paragraph in the introduction has been deleted.

5. We Supplement the  references to develop a robust and a clear conceptual framework. Please check references in the revised version.

6. Domestic cities refer to Nanjing, Chongqing, Shanghai, Suzhou, Beijing, Xi’an, Guiyang, Yangzhou, Lanzhou, Kunming, Wuhan, Urumqi and Lhasa, while overseas cities include Melbourne, London, Tokyo, Seoul, Gottingen, New York and Paris.

7. Interview data analysis process:1. we classify the interview records according to the interview outline. 2. We extract key information from the interview content for each question and code it. 3. We Analyze the encoded content and sort out the relevant frequency. 4. Get the conclusion of the interview.

8. In  phase 1 part, we try to verify and explore the effects from Ld to Lo, and further to investigate whether these effects can be divided into hygiene and motivator parts. Since hygiene (or motivator) factor discriminates target concept from, rather than negative to positive, but negative to non- negative (or positive to non-positive) parts, the analysis procedure in this part is as follows. Referring to hygiene factor, we set the score 1, 2 as negative, and 3, 4, 5 as non-negative, in Likert’s five-point scale, and then explore the discrimination of comfortableness and attractiveness on overall image through feature ranking method. Referring to motivator factor, we set the score 4, 5 as positive, and 1, 2, 3 as non-positive, and then use the same method to investigate the discrimination of these two factors on overall image.

Feature Ranking of Ld

Feature ranking usually ranks features based on their abilities to discriminate target concept (Zhou, 2016). In this study, comfortableness and attractiveness are set as the factors/features (label as Fcf and Far, respectively), and overall image as the target concept (label as To). Since information gain (IG) is widely used to characterize the amount of information that the features may contain for discriminating the target concept, we employ information gain values here to rank the discrimination importance of the features to their target concept respectively.

Taking Fcf as an example. Information gain values describe the information of Fcf ‘gained’ by splitting the data set according to its different values. where Vcf is a set of possible values that Fcf can take, and Sv is a subset of S, in which the Fcf takes the value v for all the instances, i.e.,. Information gain is actually the reduction of entropy if the data set is split using Fcf. The more reduction, the less the entropy of the resulting subset, and hence the purer in terms of the concept within each subset. That is, Fcf helps more on distinguishing between concepts.

In this study, the gain threshold is set as 0.4, according to Schaffer, Whitley and Eshelman (1992). This indicates that the discrimination importance of the features that either reaches or exceeds 0.4 could be considered to have great importance on To. The higher the value is, the greater the discrimination importance could be. There would be no importance when the value reaches 0.

The results indicate that both Fcf (IGcf_h = 0.853) and Far (IGar_m=0.772) have discrimination importance on To. Based on it, the existence of Ld, as well as the effects from Ld to Lo, are verified. This indicates that overall destination image is formed through the interaction of two characteristics, i.e., comfortableness and attractiveness, of a destination, from tourists’ perceptions.

Identifying hygiene vs. motivator factors

Based on the values of information gain, Fcf has great importance on discriminating To from negative to non-negative (IGcf_h=0.853), rather than from positive to non-positive (IGcf_m=0.321), and therefore can be treated as hygiene factor of To_h. This indicates that tourists who feel that the destination is uncomfortable, no matter physiologically or psychologically, would have negative perceptions of it; whereas if they feel that the destination is comfortable, it doesn’t mean that they would have positive perceptions. On the contrary, Far has great importance on discriminating To from positive to non-positive (IGar_m=0.772), rather than from negative to non-negative (IGar_h=0.259), and therefore can be treated as motivator factor of To_m. This means that tourists who feel that the destination is attractive would have positive perceptions of it; whereas if they feel that the destination is unattractive, it doesn’t mean that they would have negative perceptions.

9.Thank you very much for your suggestions. We think it would be better to combine 2 and 3. we agree with your advice. Appreciate you again!

All your suggestions have been revised in the paper, please check it. Thank you very much.

Reviewer 2 Report

The authors' full information including their email addresses and affiliation needs to be addressed.

The literature review should be much expanded with more discussion on each of the theoretical principles introduced.

The references style in the text should be changed to the MDPI style.

More numbers and more updated references should be cited in the literature review.

How those interviewees were approached and selected need to be explained more in detail. 

The conclusion should have different segments including 5.1 Theoretical contributions, 5.2 Managerial implications, and 5.3 Limitations and further research.

The references list at the ned of the manuscript has to be massively changed to MDPI style. 

Author Response

Thanks for your suggestions, the following is my response :

1.The complete information of the author has been supplemented, please check the latest paper.

2. We Supplement the  references to develop a robust and a clear conceptual framework. Please check references in the revised version.

3.The references style in the paper have been changed to the MDPI style.

4. The literature review format has been revised. Please check references in the revised version.

5. The questionnaire survey was carried out in October, 2019, which is the most favorite month of the year for the majority of tourists, due to the agreeable weather and national golden week holidays of China. The questionnaires were distributed to the participants at several areas, including airports, railway stations, and famous tourism sites (i.e., Sun Yat-sen’s Mausoleum, Confucius Temple, Presidential Palace, and Xuanwu Lake Park,), using a convenience sampling method. We judged the age level of the interviewees based on their appearance, and asked them if they had traveled to other cities in the recent period. If the answer is yes, we will send out further questionnaires, otherwise we will not send out. Six trained graduate students were recruited to carry out the survey. A small souvenir was given to each participant so as to express investigators’ gratitude.

6. 

Theoretical contribution

The theoretical contribution of this study is to explore destination image formation from a brand-new perspective, i.e., focusing on the interactions of perceived elements. As a result, the hierarchical fusion process of destination image formation is first detected and identified in tourism research. Methodologically, machine learning method, as the core of artificial intelligence, is applied in this study, which provides exemplary work for intelligent data analysis in quantitative tourism research.

Management significance

The findings provide practical insight into brand image construction of urban tourism destination, in a more comprehensive and targeted way. When building the brand image of urban tourism destinations, managers should not only consider a single element in the process of destination image formation, but should explore how these elements interact, and this process is systematic and progressive. Brand image building can be regarded as a dynamic integration process within the pyramid hierarchy, with the elements being from more to less, separate to compound, and concrete to abstract.

Limitations and prospections

This study has limitations as well that would be conducted in our future work. First, the data used in this study are all collected from on-site survey, while online data (such as data in travel blog or online social media) would be further used, due to the superiority of hugeness and objectivity. Second, since the main purpose of this study is to verify the existence of interactive elements as well as their dynamic hierarchy in destination image formation, we select one particular urban destination as the target case. More cities, as well as more types of tourism destinations, would be targeted in our future work. Besides, the variation of travel phases would be further considered in this hierarchical fusion process so as to find out how do perceived elements interact with each other in different travel phases.

All your suggestions have been revised in the paper, please check it. Thank you very much.

Reviewer 3 Report

This paper has potential. There are some matters, however, which need to be addressed: 

  1. Abstract: The abstract is only understandable once the article has been read. I would advise to rewrite it so that it draws the reader in prior to reading the article. 
  2. Introduction: Paragraph 2 and paragraph 3 are identical. 
  3. Sub-chapters 2.1, 2.2 and 2.3 = As the authors state, destination image is a core concept in tourism research. These sub-chapters are therefore lacking in newer (more recent) research - the seminal ones have been included (e.g., Echtner & Ritchie), yet there has been significant progress on destination image formation research since the early 1990's. A more in-depth review of literature is needed, as this could also better identify how your paper adresses any potential research gap and how it contributes to the field of destination image research (something that is also not clear in the paper). 
  4. For an international audience, I would advise to add a map of China and show Nanjing's location compared to e.g., Shanghai and Beijing, as this destination might be unfamiliar to readers. 
  5. Line 241: What are the reasons for basing your analysis on anthropomorphism? Does this resonate with previous destination image research? If not, what is the value/validity of this approach?  
  6. Figure 3: This is an important figure and presents quite a bit of information. For  better readability, I would advise to include the names in the 'boxes' (for example Fme: Modernity, and not just  Fme). 
  7. The discussion/conclusions need work: 
    • There should be more of a discussion on future research, based on your findings.
    • The conclusion seems to paraphrase the introduction and offers little to the reader. 
    • Line 573: I would disagree that there are practical insights. What do your findings mean to practitioners - what should they do based on your findings? How can DMO's better manage their destination brand, based on your findings?
    • What is the added value to both practitioners and academics with this knowledge? - answering the "so, what?" question. 
  8. References: Need to be revised, as there are several articles present in the reference list, but not in the text (e.g., Papadimitriou et al., 2015). Moreover, there are several newer papers in the reference list pertaining to destination image formation - why are they not discussed in the literature review? 

Author Response

Thanks for your suggestions, the following is my response :

1.Abstract: Image has been widely accepted as a combination of perceived elements which are commonly discrete and static ones. ‘Discrete’ means that the elements are treated as separate ones with each other, with no interactions among them. ‘Static’ means that the elements would not be changed into other forms in the process of destination image formation. This study, thinking outside the box, tries to explore destination image formation through perceived elements, taking their interactions and corresponding changes into account. Machine learning, as the core of artificial intelligence, is applied for data analysis in this study. Urban tourism destination is targeted because of its variety and abundance of perceived elements. Data are collected from both interview and questionnaire survey of tourists. Through several phases of analysis, this study finally finds that, in tourism destination image formation, perceived elements do interact with each other and change into new forms level by level. Specifically, there are at least four levels in the whole process of destination image formation, i.e., individual-landscape layer, compound-atmosphere layer, dual-factor layer, and overall-image layer, from bottom to top. In the bottom stage, elements are commonly numerous, separate, and concrete. With the interactive effects of the elements, they integrate with each other and generate some fewer new forms in higher level, which would be more general and abstract. This process goes on and on until the formation of the most general one, i.e., overall destination image. Based on the findings, the dynamic fusion process and pyramid hierarchy of destination image formation is disclosed. This study explores destination image formation from a brand-new perspective, considering perceived elements within a dynamic, synthetic system, and therefore provides practical insights into destination image construction in a more comprehensive and targeted way.

2.The same paragraph in the introduction has been deleted.

3.We Supplement the  references to develop a robust and a clear conceptual framework. Please check references in the revised version.

4.The location of Nanjing relative to Beijing and Shanghai on the map of China has been Supplemented in the paper.

5. Since human-like trait has been accepted as an effective representation of general characteristic referring to a particular object (Aaker, 1997), based on the theory of anthropomorphism, we asked participants to think of the city as a person, and to estimate the matching rate of it with a pool of descriptive adjectives, ranging from 1(not match at all) to 5 (greatly match). This resonates with previous studies of destination imagery. Based on the theory of anthropomorphism, the items were collected in two ways, i.e., urban destination personality scale (Zhang et al., 2019), and interview survey that were used to supplement for a specific case.

6. The complete information of the figure has been added in the paper. Please see the attachment.

7.

Limitations and prospection

This study has limitations as well that would be conducted in our future work. First, the data used in this study are all collected from on-site survey, while online data (such as data in travel blog or online social media) would be further used, due to the superiority of hugeness and objectivity. Second, since the main purpose of this study is to verify the existence of interactive elements as well as their dynamic hierarchy in destination image formation, we select one particular urban destination as the target case. More cities, as well as more types of tourism destinations, would be targeted in our future work. Besides, the variation of travel phases would be further considered in this hierarchical fusion process so as to find out how do perceived elements interact with each other in different travel phases.

Conclusions

Through machine learning technique, we finally find that: perceived elements do interact with each other; with the interaction of elements in different ways, new forms of elements are generated in higher level; this process continues level by level, with the elements being from more to less, separate to compound, and concrete to abstract. Specifically, there are at least four levels in the whole process of destination image formation, i.e., individual-landscape layer, compound-atmosphere layer, dual-factor layer, and overall-image layer, from bottom to top. In the bottom stage, elements are commonly numerous, separate, and concrete. With the interactive effects of the elements, they integrate with each other and generate some fewer new forms in higher level, which would be more general and abstract. This process goes on and on until the formation of the most general one, i.e., overall destination image. Based on the findings, the dynamic fusion process and pyramid hierarchy of destination image formation is disclosed. This study explores destination image formation from a brand-new perspective, considering perceived elements within a dynamic, synthetic system, and therefore provides practical insights into destination image construction in a more comprehensive and targeted way.

from the perspective of practitioners, When building the brand image of urban tourism destinations, managers should not only consider a single element in the process of destination image formation, but should explore how these elements interact, and this process is systematic and progressive. Brand image building can be regarded as a dynamic integration process within the pyramid hierarchy, with the elements being from more to less, separate to compound, and concrete to abstract. 

For scholars, this study is to explore destination image formation from a brand-new perspective, i.e., focusing on the interactions of perceived elements. As a result, the hierarchical fusion process of destination image formation is first detected and identified in tourism research. Methodologically, machine learning method, as the core of artificial intelligence, is applied in this study, which provides exemplary work for intelligent data analysis in quantitative tourism research.

8.The literature review  has been revised. Please check references in the revised version.

All your suggestions have been revised in the paper, please see the attachment. Thank you very much.

Round 2

Reviewer 2 Report

The revised paper looks good with successful answers to the reviewers' comments.  Please check and make minor English editing.

Author Response

Thanks for your suggestions, the following is my response :

I have checked and made minor English editing. The changes in the paper are marked in red. All your suggestions have been revised in the paper, please check it. Thank you very much.

Reviewer 3 Report

Much improved paper after revision - I only have one comment: The map of China should be in English - as this is particularly helpful for the international audience. 

Author Response

Thanks for your suggestions, the following is my response :

The map of China have been revised in English. Besides, I have checked and made minor English editing. The changes in the paper are marked in red. All your suggestions have been revised in the paper, please check it. Thank you very much.
